# Fabrication of Ultra-Fine and Ultra-Long Copper Tube Electrodes by Ultrasonic High-Frequency Percussion

**DOI:** 10.3390/mi13091405

**Published:** 2022-08-27

**Authors:** Xiajunyu Zhang, Yugang Zhao, Hanlin Yu, Zhihao Li, Chuang Zhao, Guangxin Liu, Chen Cao, Qian Liu, Zhilong Zheng, Dandan Zhao

**Affiliations:** School of Mechanical Engineering, Shandong University of Technology, Zibo 255049, China

**Keywords:** ultrasonic high-frequency percussion, ultrasonic high-frequency percussion equipment, ultra-fine and ultra-long copper tube electrode, surface roughness, core leach

## Abstract

In this study, a new method of ultrasonic high-frequency percussion (UH-FP) is proposed. Ultra-fine and ultra-long copper tube electrodes cannot be fabricated by traditional processing methods, and the copper tube electrodes fabricated by UH-FP can be used in the process of rotary EDM for microfine holes. The UH-FP setup has been established based on an ultrasonic device, a workpiece chucking and rotation device, and a workpiece reciprocating motion device. In this work, by studying the principle of ultrasonic processing, the processing principle and mechanism of ultra-fine and ultra-long copper tube electrode preparation by ultrasonic high-frequency percussion is proposed. The effects of processing parameters (i.e., rotational speed, feed rate, working gap, percussion amplitude) on surface roughness are evaluated quantitatively. Experimental results show that the proposed method could complete the core leach of the core-containing copper tube electrodes after drawing, while improving surface quality. Some surface defects such as cracks, scratches and folds were completed removed, further improving the mechanical performance of processed parts. The surface roughness (Ra) of 0.091 μm was obtained from the initial 0.46 μm under the optimal processing parameters of 800 rpm tube rotational speed, 200 mm/min platform feed speed, 0.13 mm machining gap, 0.15 mm percussion amplitude, and 32 min machining time. The method shows potential for manufacturing copper tube electrodes for a wide range of industrial applications.

## 1. Introduction

Micro holes have widespread applications in aerospace, precision equipment, and automotive manufacturing [1]. Currently, there are many methods for manufacturing micro holes, including electrical discharge machining (EDM). EDM has been widely used for micro-hole processing because of its non-contact nature, low residual stress, low cost, and high economic efficiency [2]. Material is removed from the workpiece surface with the conversion of electrical energy to thermal energy. Plasma channels are formed between the machining tool and the workpiece surface at certain voltages. Material is removed from the workpiece surface in non-solid form, with the conversion of electrical energy to thermal energy. EDM has extremely low tool loss compared to conventional machining methods [3]. In addition to the influence of different machining parameters on EDM-machining quality, the performance of the electrodes for EDM is one of the most important factors affecting machining quality. However, the problem of chip evacuation during machining weakens the machining quality of EDM. A rotatable tubular electrode was utilized to improve this problem, EDM-machined holes having better machining quality compared to those using normal fixed electrode [4]. Currently, machining methods used for electrode include block electrode discharge grinding (BEDG), wire electrical discharge grinding (WEDG), electrochemical etching, etc. [5,6,7,8,9]. Research on the fabrication of tubular electrodes has been relatively rare. The traditional processing of tube electrodes can be divided into the following steps. Firstly, coring and drawing for tubular electrodes; then, decoring by low-frequency mechanical percussion; finally, post-processing. Although the traditional method is effective for fabricating tubular electrodes, the fabrication process is tedious. As electrodes are becoming more demanding in quality and finer in diameter, low-frequency mechanical percussion is difficult to use to complete the processing task of core leach, and can even make the surface of the fabricated electrodes defective [10]. Meanwhile, ultrasonic processing has the characteristics of high processing frequency, small macroscopic force, and good surface quality of the workpiece after processing. Especially in the field of difficult-to-machine materials, the method has solved many process problems, thus indicating that ultrasonic processing can be applied to solve key technical problems that hinder decoring by mechanical percussion in traditional methods [11]. However, few studies hae been carried out on the fabrication methods of tubular electrodes with ultrasonic technology as the mainstay.

To fill this gap, this study proposes an ultrasonic high-frequency percussion (UH-FP) method for the fabrication of ultra-fine and ultra-long copper tube electrodes. Firstly, the processing principles and mechanism of fabricating ultra-fine and ultra-long copper tube electrodes by ultrasonic high-frequency percussion was revealed by exploring various ultrasonic-related processing principles and methods, which provided the theoretical basis for the subsequent experimental processing. Secondly, the experimental equipment was developed for processing the ultra-fine and ultra-long copper tube electrodes fabricated by ultrasonic high-frequency percussion. Finally, single-factor experiments were designed and conducted. The effect of each processing parameter on the surface roughness of the outer surface of the tube was quantitatively analyzed. From the analysis of the experimental results, the combination of processing parameters suitable for this experimental tube was obtained, and the processing results with this combination of processing parameters were found to be very satisfactory.

## 2. Experimental Setup and Materials

### 2.1. Experimental Setup

Figure 1 shows the developed experimental equipment for processing ultra-fine and ultra-long copper tube electrodes fabricated by ultrasonic high-frequency percussion. Figure 2 shows the control system for ultrasonic high-frequency percussion fabrication of ultra-fine and ultra-long copper tube electrode equipment. The mechanical structure of the equipment mainly consisted of three parts (e.g., ultrasonic device, workpiece chucking and rotation device, and workpiece reciprocating motion device). The workpiece clamping and rotation device mainly consisted of an AC servo motor, tensioning device, and precision chuck. The tube was chucked by the precision chuck and tensioned by the tensioning device, rotating by the AC servo motor, thus achieving the rotational movement of the workpiece. The workpiece reciprocating motion device was composed of a stepper motor, synchronous belt linear module, machining platform, and limit switch. Under the drive of the stepping motor, the synchronous belt linear module drives the machining platform to make a reciprocating motion, which can realize the machining of whole sections of tubes. The ultrasonic device included an ultrasonic generation section and an ultrasonic positioning section; the ultrasonic generation device was composed of a transducer, a variable amplitude rod, an ultrasonic forging device, and a buffing device; the ultrasonic positioning device was composed of an AC servo motor, a ball-screw linear module, and a limit switch. The AC servo motor drives the ball screw linear module to move linearly in the direction perpendicular to the processing plane. The ultrasonic generation device was mounted on the ball-screw linear module, which realized the positioning between the ultrasonic forging device and the tube. Also, the rear holder can be moved on the base to enable the processing of different lengths of tubes by adjusting the distance between it and the front holder. The interface of the control system was written in the Delphi language and the control system program was written in VC++6.0. The combined control of AC servo motors installed symmetrically in the same axis, stepping motors in the horizontal direction, and AC servo motors in the vertical direction can be realized simultaneously, to make the tube rotate and reciprocate, and the ultrasonic forging device move linearly perpendicular to the processing plane. The adjustable range of the experimental equipment for ultrasonic percussion processing of copper tube electrodes is shown in Table 1.

### 2.2. Copper Tube Electrodes and Its Performance Parameters

The experimental copper tube electrode is shown in Figure 3. The material is copper, the wire core material is grade 45 steel, the length of copper tube is 600~900 mm, the inner diameter is 1.20 mm and the outer diameter is 1.24 mm. The specific elemental composition is shown in Table 2 [12]. The tube performance parameters are shown in Table 3 [13].

## 3. The Principle and Mechanism of Fabricating Copper Tube Electrodes by Ultrasonic High-Frequency Percussion

Figure 4 shows the principle of fabricating ultra-fine and ultra-long copper tube electrodes by ultrasonic high-frequency percussion. The core-containing copper tube electrode is chucked on both sides of the motor by the precision chuck, during coaxial rotation movement driven by both sides of the motor, while the synchronous belt drives the platform as a whole to complete the reciprocal translation movement. The high-frequency ultrasonic forging device targets constant percussion to the copper tube electrode, so that plastic deformation the copper tube electrode occurs under the effect of continuous high-frequency percussion. Therefore, the surface of the copper tube electrode is reinforced, and the copper tube electrode from the wire core loosens and detaches, and finally the processing of the copper tube electrode is complete. Figure 5 shows the processing mechanism of ultrasonic high-frequency percussion for the fabrication of ultra-fine and ultra-long copper tube electrodes. The core-containing copper tube electrode moves along the processing direction at certain speed, and the ultrasonic forging device continuously performs high-frequency percussion on the copper tube electrode at a suitable machining gap. The force acting on the copper tube electrode can cause only a small plastic deformation of the tube, because ultrasonic percussion has the property of no macroscopic force. The tube is squeezed in the direction of processing under the constraint of the wire core. As the process progresses, the high frequency vibration generated by the percussion causes the copper tube electrode to loosen from the wire core, the tube wall to be thinned to some extent, and eventually the copper tube electrode to detach completely from the wire core. Meanwhile, ultrasonic high-frequency percussion causes the surface of copper tube electrode to produce reinforcement and achieve a certain brightening effect, which reduces the surface roughness of copper tube electrode and removes the cracks, scratches, folds, and other defects from the original surface of the tube.

Figure 6 shows the different machining conditions in the UH-FP process, which have an important impact on the machining results and directly determine the successful core leach of copper tube electrodes.

Figure 6a shows the state when the machining gap is larger than the percussion amplitude. In this state, no matter how the ultrasonic forging device is percussed, the high frequency percussion force cannot be applied to the surface of the tube, and the tube cannot be processed for core leach.

Figure 6b shows the state when the machining gap is equal to or slightly less than (slightly greater than) the percussion amplitude. The ultrasonic forging device driven by ultrasonic waves continuously exerts a small high-frequency percussion force on the tube, and through the relative movement of the ultrasonic forging device and the tube, core leach and surface finishing are achieved. At this stage, the force for the core leach and surface finishing is derived from the high-frequency percussion force of the ultrasonic forging device, under ideal processing conditions.

Figure 6c shows the state when the processing gap is much smaller than the percussion amplitude. While applying high-frequency percussive force, the ultrasonic forging device makes full contact with the tube, resulting in excessive force. New defects such as percussion dents and scratches appear on the surface after percussion, and the tube vibrates irregularly with percussion, which affects the processing effect. In this case, although core leach of the tube can be completed, it is difficult to obtain good processing quality or higher surface quality, and the tube may even be broken or frequently chipped.

Figure 7 shows the trajectory of the percussive motion of the ultrasonic forging device acting on the outer surface of the tube. During the machining process, the tube rotates under the drive of servo motor and the machining platform is reciprocated axially under the traction of the stepper motor. Thus, the ultrasonic forging device acts on the outer surface of the tube as a helix formed by the rotation of the tube itself and the axial reciprocating motion of the platform. By adjusting the control system, the tube is rotated clockwise when the platform moves axially in the positive direction, and the tube continues to rotate clockwise when the platform moves axially in the negative direction, so that the percussive motion trajectory changes to a cross spiral, and the percussive motion trajectories at the intersection are perpendicular to each other. This machining state allows more comprehensive coverage of the machining trajectory and produces better machining results. The velocity *V* of this trajectory is synthesized from the rotational speed *V_x_* and the axial feed speed *V*_y_, which is given by the following equation:$$V=\sqrt{{V_{x}}^{2}+{V_{y}}^{2}}=\sqrt{{\left(\frac{\pi \mathrm{nd}}{60}\right)}^{2}+{V_{y}}^{2}}$$
where *V_x_* is the rotational speed, mm/s; *V_y_* is the axial feed speed, mm/s; n is the workpiece speed, rpm; d is the diameter of the workpiece, mm.

The density of the percussive motion trajectory is mainly determined by two process parameters: the tube speed and the platform’s axial feed speed. When the axial feed speed of the platform remained constant and the tube speed was reduced, the percussive motion trajectory was relatively sparse and the number of machining times in the adjacent area was too few, resulting in lower machining effect and longer machining time. With the gradual increase of tube speed, the percussive motion trajectory becomes more and more intensive, the number of adjacent areas processed increased, the processing effect was significantly improved, the percussion was more uniform, the rate of core leach increased, and the surface quality was superior. However, when the tube speed is too high, the adjacent area was processed too many times, leaving the tube unable to withstand the impact and breakage, resulting in processing failure.

When the tube speed was kept constant and the platform axial feed speed was low, the percussive motion trajectory was dense, and the slower the feed speed, the denser the resulting percussive motion trajectory. As the platform axial feed speed increased, the percussive motion trajectory became sparser.

## 4. Experimental Results and Discussion

A single-factor experiment was designed to investigate the effects of tube rotational speed, platform feed speed, machining gap, percussion amplitude, and machining time on the surface roughness of the outer surface of the copper tube electrode.

A small amount of lubricant was applied evenly on the surface of the core-containing copper tube electrode, the wire core with precision chucks at both ends was tightened. The platform was moved and the tube position adjusted to the start point of the processing. The ultrasonic forging device was moved to find the contact point between ultrasonic forging device and tube, that place was set as zero point and input into the computer program, then the ultrasonic high-frequency percussion processing experiment was carried out. After the experiment was completed, the copper tube electrode was removed from the wire core, an appropriate amount of anhydrous ethanol was added to the beaker and the copper tube electrode was placed into the ultrasonic cleaner. Ultrasonic cleaning was applied to avoid stains from subsequent observation and measurements of the surface of the tube, which may otherwise have affected the results. Five positions were selected for each experimental tube, spaced 3 mm apart, and the surface roughness of each selected position of the tube was measured using a DSX1000 3D digital microscope (DSX1000, OLYMPUS, Tokyo, Japan) and the average value calculated.

### 4.1. Effect of Tube Rotational Speed on Surface Roughness

The experimental conditions selected for different tube rotational speeds are shown in Table 4. Figure 8 shows the effect of tube rotational speed on the surface roughness of the copper tube electrode.

As can be seen from the graph, the processing effect was more significant when the tube rotational speed was below 1000 rpm, and the surface roughness obtained was more desirable when the tube rotational speed was less than 1000 rpm, all at less than 0.2 μm. When the tube rotational speed was 800 rpm, the surface roughness was the smallest, at 0.106 μm. When the tube rotational speed was greater than 1000 rpm, the surface roughness tended to increase. This is because the tube rotational speed was too high; at any position, the ultrasonic forging device had not yet made fully contact with the tube, and the tube has already rotated to the next position, resulting in incomplete processing. Although the tube core was successfully leached, it did not achieve the optimal effect of surface strengthening and finishing.

### 4.2. The Effect of Platform Feed Speed on Surface Roughness

The experimental conditions selected with different platform feed speeds are shown in Table 5. Figure 9 shows the effect of platform feed speed on the surface roughness of copper tube electrode.

As can be seen from the figure, when the platform feed speed was 200 mm/min, the surface roughness obtained was the smallest at 0.135 μm. Thereafter, the surface roughness gradually increased as the platform feed speed continuously increased or decreased. When other conditions remained unchanged, an increase in the platform feed speed leaves the same position subject to percussion for a shorter period of time and prevents full percussion processing, resulting in the tube being fed to the next position before plastic deformation occurs, affecting the quality of the machined tube surface. The faster the platform feed speed, the more pronounced this phenomenon was found to be, even affecting core leach. When all other conditions remained the same and the platform feed speed decreased, the same position was subject to increased percussion time, and the tube was subject to excessive percussion processing, bringing new processing marks and affecting the quality of the machined tube surface. The slower the platform feed speed, the more pronounced this phenomenon became, even breaking the tube.

### 4.3. Effect of Machining Gap on Surface Roughness

The experimental conditions selected at different machining gaps are shown in Table 6. Figure 10 shows the effect of machining gap on the surface roughness of copper tube electrode.

As can be seen from the figure, the surface roughness always remained at a high value, above 0.25 μm, when the machining gap was relatively small; as the machining gap increased, the surface roughness decreased. This was caused by the machining gap being excessively small compared with the percussion amplitude. As shown in Figure 6c, when the machining gap was too small for the percussion amplitude, there was increased contact between the ultrasonic forging device and the tube, resulting in excessive force. Although the tube core were eventually leached, the excessive processing resulted in new defects on the tube surface, resulting in high surface roughness values and poor surface quality. When the machining gap was above 0.125 mm, the surface roughness obtained was closer to the ideal, less than 0.16 μm. The smallest surface roughness of 0.121 μm was obtained when the machining gap was 0.13 mm. When the machining gap was greater than 0.13 mm, the surface roughness showed a tendency to increase, but the rise was not obvious.

### 4.4. The Effect of Percussion Amplitude on Surface Roughness

The experimental conditions selected for different percussion amplitudes are shown in Table 7. Figure 11 shows the effect of percussion amplitude on the surface roughness of the copper tube electrode.

As can be seen from the figure, the surface roughness was not ideal in the case of small percussion amplitude, at above 0.3 μm. Although the core leach was completed, the surface quality was poor and the desired surface strengthening and finishing was not achieved. As shown in Figure 6b, although the machining gap was equal to or slightly larger (slightly smaller) than the percussion amplitude, the machining process was carried out normally. Because the difference between the two was too small, the force did not fully act on the surface of the tube, and more machining time was needed to achieve the surface strengthening and finishing effect. With the increased percussion amplitude, the force acting on the surface of the tube gradually increased and the surface roughness was reduced. When the percussion amplitude was 0.15 mm, the surface roughness was the smallest, at 0.117 μm; when the percussion amplitude increased further, the surface roughness also showed an increasing trend.

### 4.5. Effect of Machining Time on Surface Roughness

The experimental conditions selected for different machining times are shown in Table 8. Figure 12 shows the effect of machining time on the surface roughness of the copper tube electrode.

From the figure, it can be seen that the longer the machining time within a certain degree, the smaller the surface roughness, but when the time was too long, the surface roughness started to increase gradually, showing a general trend of first decreasing and then increasing. Surface roughness reached a minimum at a machining time of 32 min, at 0.11 μm. The surface roughness then tended to increase, due to the fact that with the increase of machining times, new machining marks appeared on parts of the surface that had been reinforced and brightened by repeated percussion from the ultrasonic forging device, resulting in an increase of surface roughness.

### 4.6. Comparison of Surface Morphology before and after Machining

Figure 13a shows a 3D digital micrograph of the outer surface of the tube before machining. At this point, the roughness of the outer surface of the tube was 0.46 μm. From the 3D digital micrograph, cracks, scratches, folds, and other defects can be seen on the surface of the tube before the percussion. These defects would seriously affect the processing effect of the copper tube electrode in the EDM, causing deterioration in the surface quality of the machined workpiece. Therefore, these defects must be removed.

Figure 13b shows the 3D digital micrograph after 32 min of percussion, with selected machining parameters of tube rotational speed 800 rpm, platform feed speed 200 mm/min, machining gap 0.13 mm, and percussion amplitude 0.15 mm. At this stage, the surface roughness of the tube surface was 0.091 μm. In addition, the 3D digital micrographs from after the percussion process clearly show that all the defects on the tube surface were removed, and the surface morphology was flat and uniform, which indicates that the effect of the ultrasonic high-frequency percussion was ideal.

From the above analysis, it is clear that ultrasonic high-frequency percussion treatment of core-containing copper tube electrodes can complete the core leach and remove the surface defects of the material itself, with significant surface strengthening and finishing effect, and that the surface roughness can be reduced to less than 0.1 μm.

## 5. Conclusions

Based on the experimental results of ultrasonic high-frequency percussion of copper tube electrodes, the following conclusions are drawn.

In this study, ultrasonic high-frequency percussion processing was utilized to fabricate core-containing ultra-fine and ultra-long copper tube electrodes. The core leach was completed, and cracks, scratches, folds, and other defects were removed from the surface to achieve enhanced surface finishing.

Through single-factor experiments, the effects of tube rotational speed, platform feed speed, machining gap, percussion amplitude, and machining time on the surface roughness of the copper tube electrode were investigated, achieving a reduction in the surface roughness of the tube from an initial 0.46 μm to 0.091 μm, providing a basis for subsequent research.

The surface roughness of the copper tube electrode decreased to 0.091 μm from its initial 0.46 μm under optimal processing parameters, i.e., 800 rpm tube rotational speed, 200 mm/min platform feed speed, 0.13 mm machining gap, 0.15 mm percussion amplitude, and 32 min of machining time, providing the basis for follow-up work.

## Figures and Tables

**Figure 1 micromachines-13-01405-f001:**
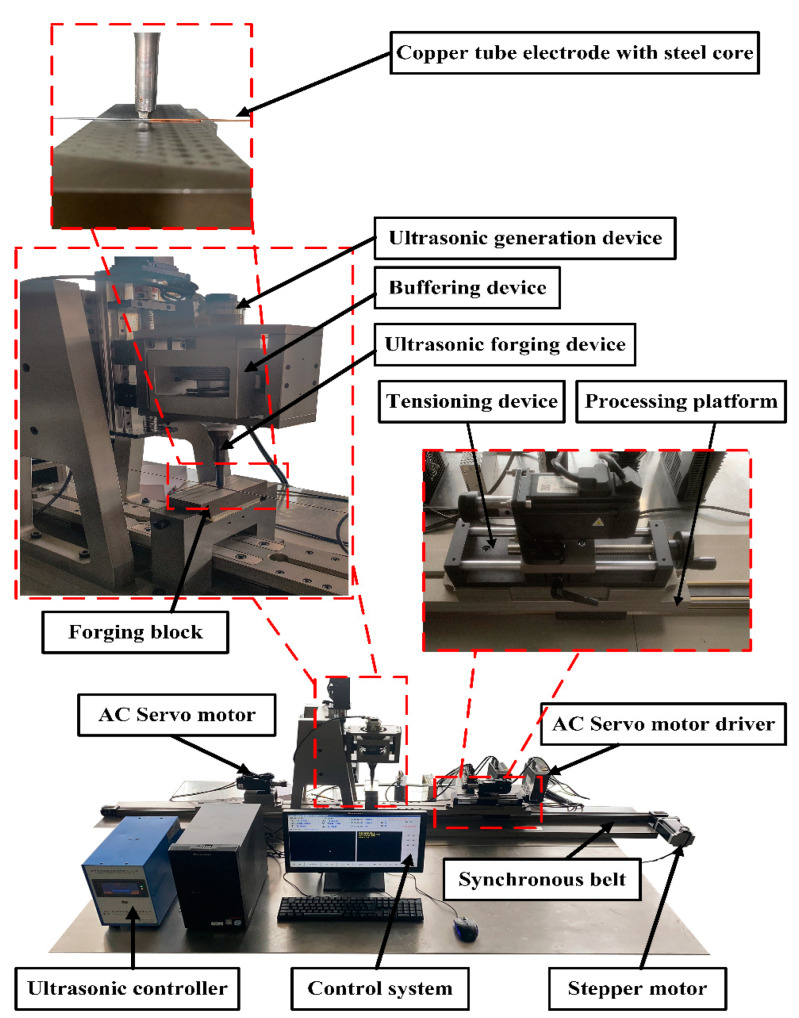
Ultrasonic high-frequency percussion fabrication of ultra-fine and ultra-long copper tube electrode experimental processing equipment.

**Figure 2 micromachines-13-01405-f002:**
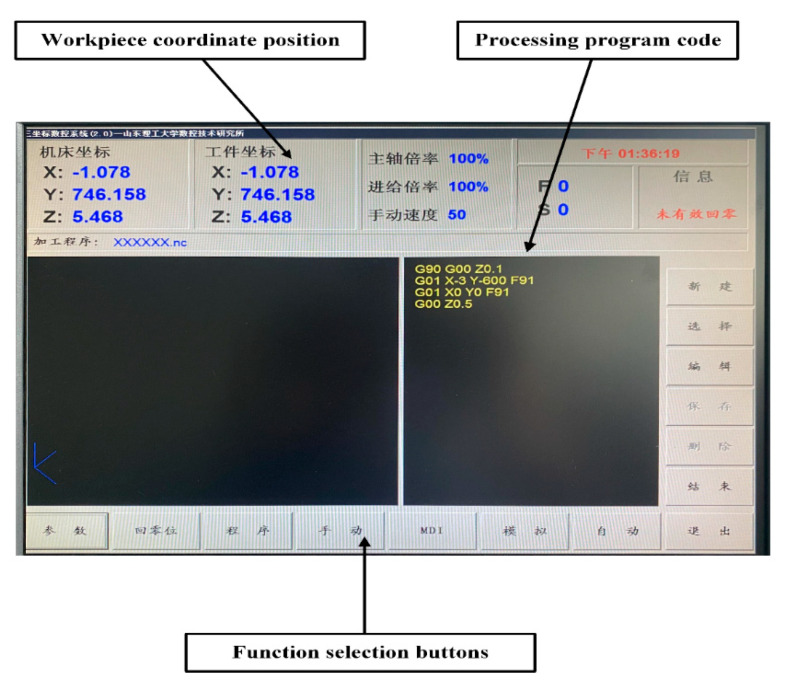
Interface of the control system for ultrasonic high-frequency percussion fabrication of ultra-fine and ultra-long copper tube electrode equipment.

**Figure 3 micromachines-13-01405-f003:**
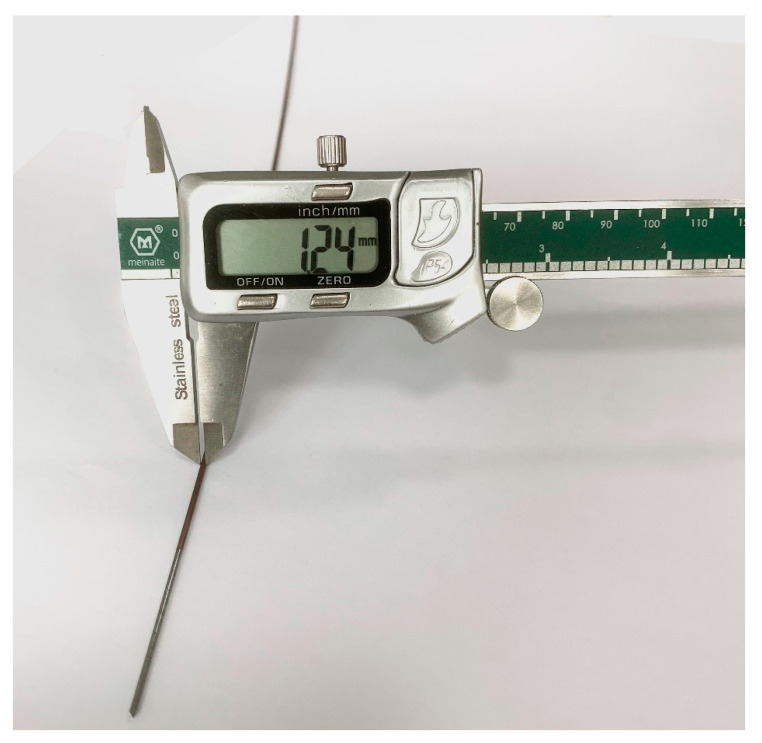
Experimental core-containing copper tube electrode.

**Figure 4 micromachines-13-01405-f004:**
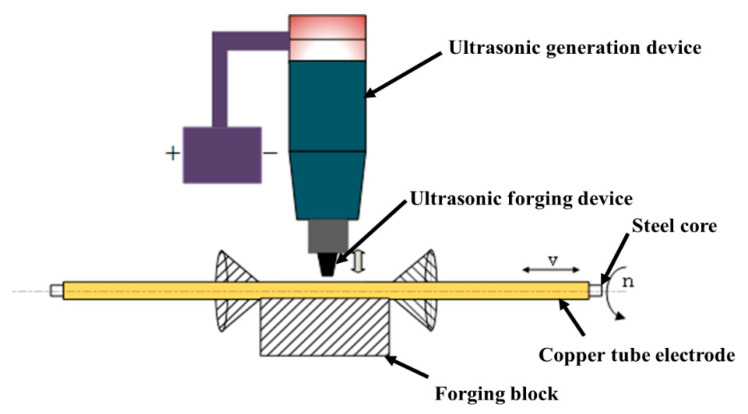
Principle diagram of ultrasonic high-frequency percussion fabrication of ultra-fine and ultra-long copper tube electrodes.

**Figure 5 micromachines-13-01405-f005:**
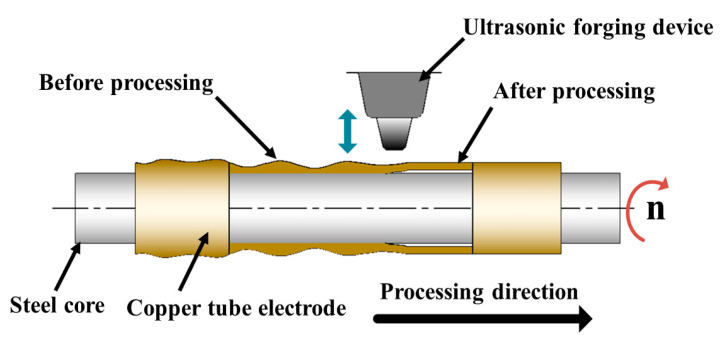
Ultrasonic high-frequency percussion fabrication of the processing mechanism for ultra-fine and ultra-long copper tube electrode.

**Figure 6 micromachines-13-01405-f006:**
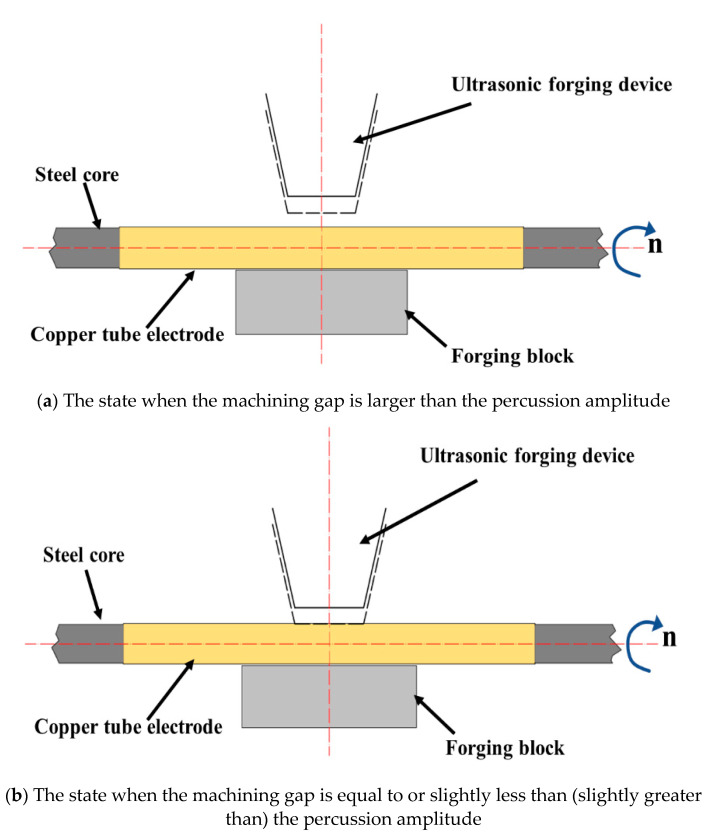
Machining conditions resulting from different machining clearances.

**Figure 7 micromachines-13-01405-f007:**
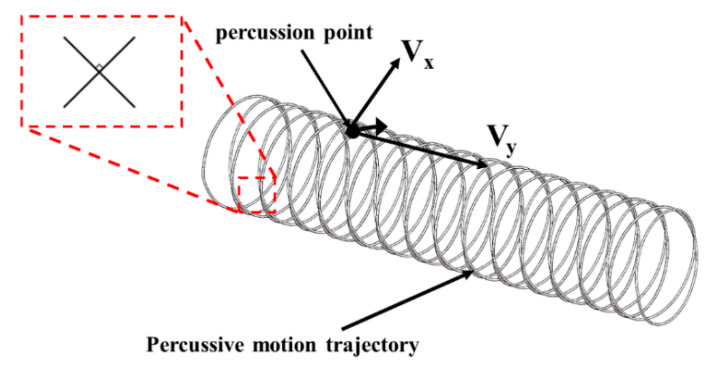
Motion trajectory of percussion point.

**Figure 8 micromachines-13-01405-f008:**
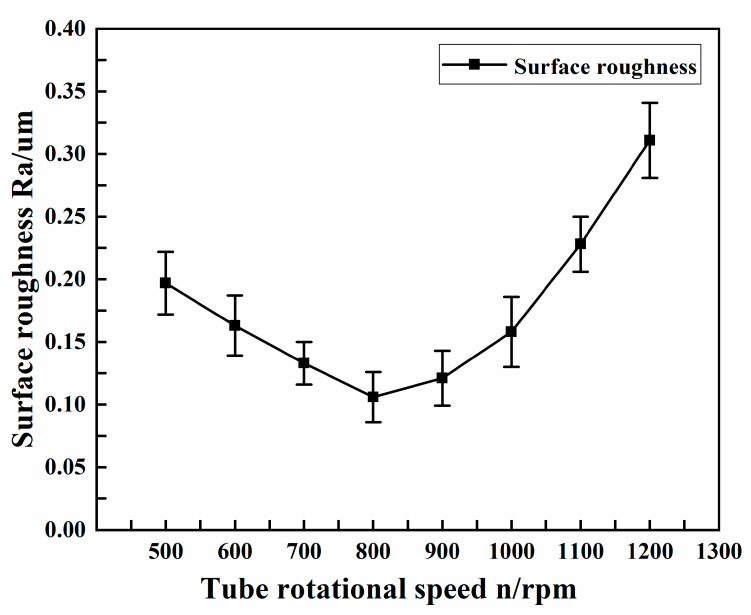
Effect of tube rotational speed on surface roughness.

**Figure 9 micromachines-13-01405-f009:**
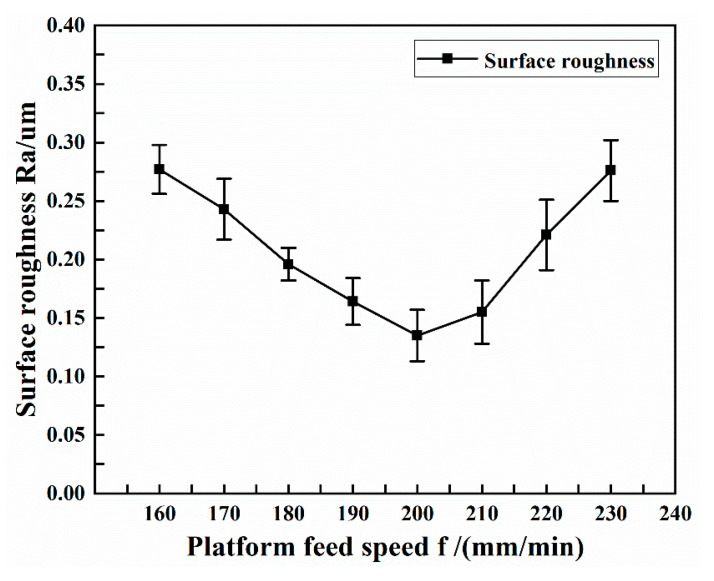
Effect of platform feed speed on surface roughness.

**Figure 10 micromachines-13-01405-f010:**
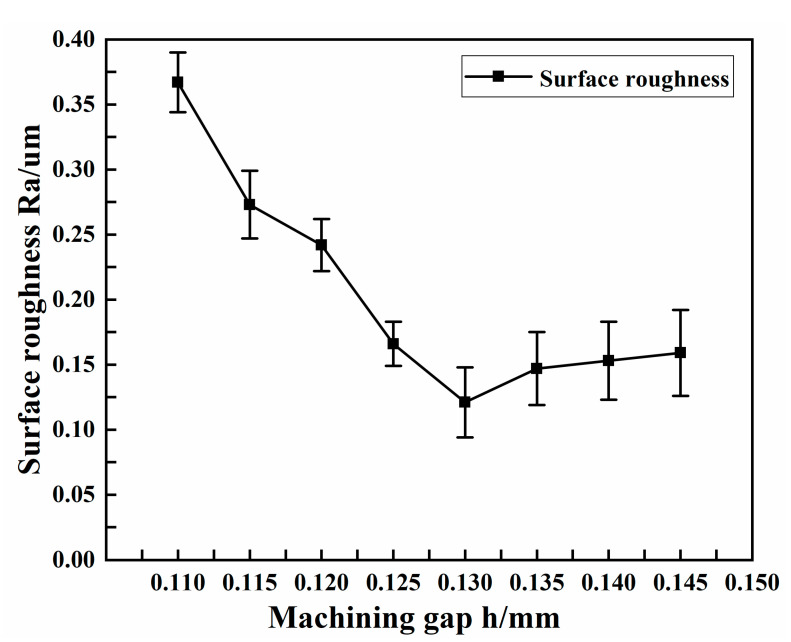
Effect of machining gap on surface roughness.

**Figure 11 micromachines-13-01405-f011:**
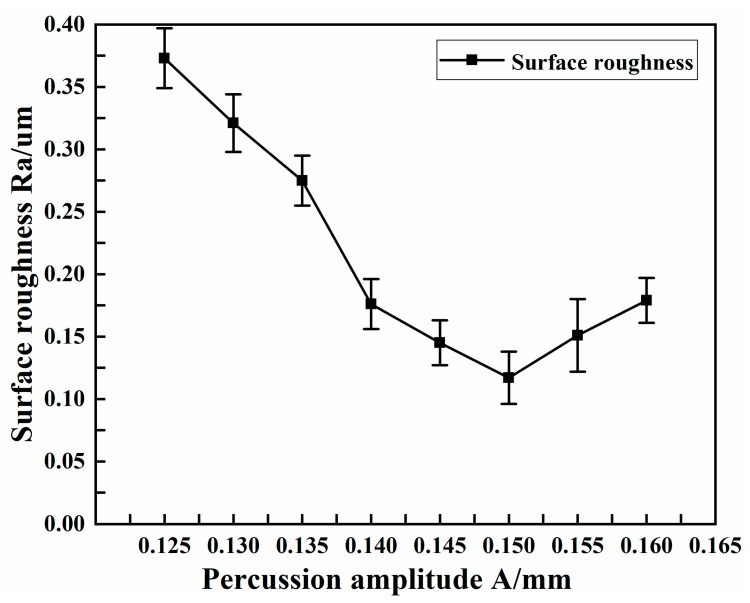
Effect of percussion amplitude on surface roughness.

**Figure 12 micromachines-13-01405-f012:**
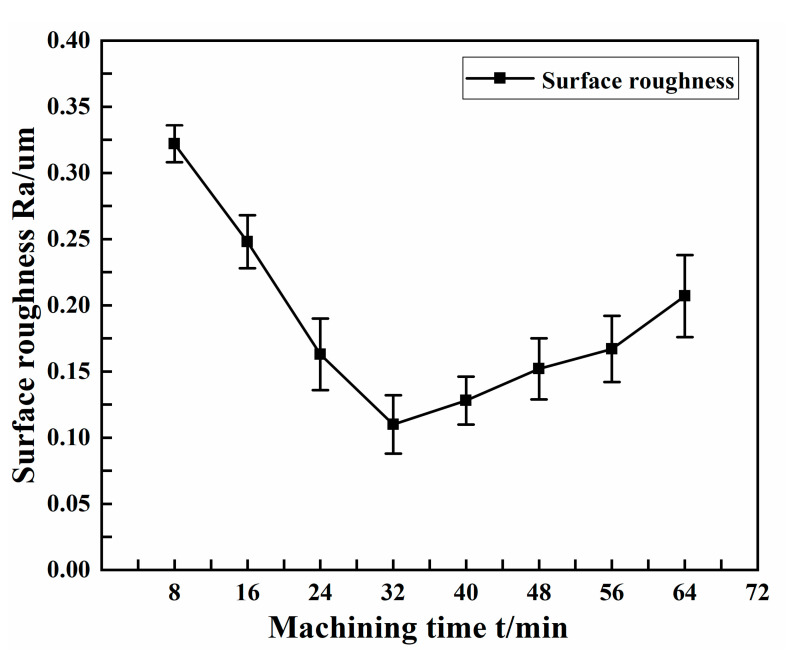
Effect of machining time on surface roughness.

**Figure 13 micromachines-13-01405-f013:**
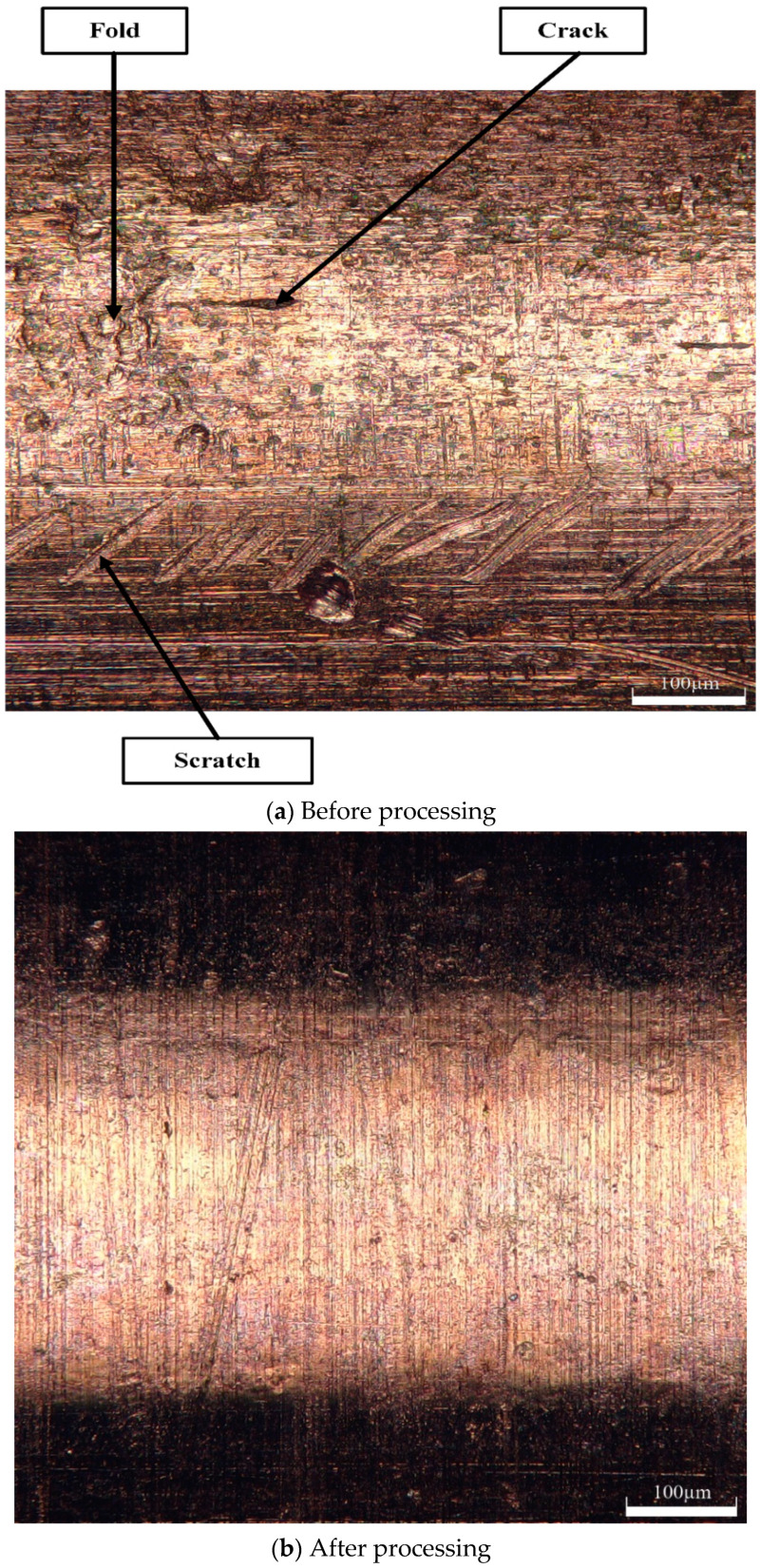
3D digital micrographs of the outer surface of the tube before and after processing.

**Table 1 micromachines-13-01405-t001:** Performance of ultrasonic percussion processing experimental equipment for copper tube electrodes.

Performance	Value
Range of processed tube diameters (mm)	0.2–1.5
Machining stroke (mm)	50~1500
Workpiece rotation speed (rpm)	0~3000
Reciprocating motion speed (mm/min)	0~300

**Table 2 micromachines-13-01405-t002:** Elemental constituents of copper tube.

Element	Cu	Bi	Sb	As	Fe	Pb	S
w/%	Bal.	≤0.001	≤0.002	≤0.002	≤0.005	≤0.005	≤0.005

**Table 3 micromachines-13-01405-t003:** Performance parameters of copper tube.

Performance Indicators	Density (g·cm^3^)	Elastic Modulus (GPa)	Tensile Strength (MPa)	Yield Strength (MPa)	Elongation (%)
Value	8.40	107.6	279.5	269.0	34.6

**Table 4 micromachines-13-01405-t004:** Experimental conditions selected for different tube rotational speeds.

Process Parameters	Value
Tube rotational speed (rpm)	500–1200
Platform feed speed (mm/min)	200
Machining gap (mm)	0.13
Percussion amplitude (mm)	0.15
Machining time (min)	32

**Table 5 micromachines-13-01405-t005:** Experimental conditions selected for different platform feed speeds.

Process Parameters	Value
Tube rotational speed (rpm)	800
Platform feed speed (mm/min)	160–230
Machining gap (mm)	0.13
Percussion amplitude (mm)	0.15
Machining time (min)	32

**Table 6 micromachines-13-01405-t006:** Experimental conditions selected for different machining gaps.

Process Parameters	Value
Tube rotational speed (rpm)	800
Platform feed speed (mm/min)	200
Machining gap (mm)	0.11–0.145
Percussion amplitude (mm)	0.15
Machining time (min)	32

**Table 7 micromachines-13-01405-t007:** Experimental conditions selected for different percussion amplitudes.

Process Parameters	Value
Tube rotational speed (rpm)	800
Platform feed speed (mm/min)	200
Machining gap (mm)	0.13
Percussion amplitude (mm)	0.125–0.16
Machining time (min)	32

**Table 8 micromachines-13-01405-t008:** Experimental conditions selected for different machining times.

Process Parameters	Value
Tube rotational speed (rpm)	800
Platform feed speed (mm/min)	200
Machining gap (mm)	0.13
Percussion amplitude (mm)	0.15
Machining time (min)	8–64

## Data Availability

No applicable.

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
