# Peer review of "Fabrication of Ultra-Fine and Ultra-Long Copper Tube Electrodes by Ultrasonic High-Frequency Percussion"

_micromachines, 2022, doi:10.3390/mi13091405_

Round 1

Reviewer 1 Report

Minor revision 

The effects of processing parameters (i.e. rotational speed, feed rate, working gap, percussion amplitude) on surface roughness were evaluated quantitatively. The cracks, scratches and folds were completed removed

1. Last Paragraph of the Introductory Section- Repairs required. 

2. Language Errors of Entire manuscript should be corrected. 

3. What happened tube rotational speed is less than 500 rpm? Whether increasing or not?

4. What happened platform feed speed is less than 160 mm/min?

5.  Annotate Figure 13 a and b.

6.  Include more surface roughness graphs (cracks, scratches and folds) to improve the explanation. 

7.  r/min -->use standard unit/ SI unit.

8.  The preset mean value of each process parameter taken for the experiments is optimum, how can you predict before the test?

9. All middle value of the each parameter is optimum? How is correct? Or / data manipulated?

10. Introduction provide sufficient background and include all relevant references. 

11. Improve abstract/ conclusion chapters.

Reviewer 2 Report

The authors have worked hard to produce the manuscript, and the presented results agree with the method presented. The following are a few of the comments that authors can work on to better highlight the impact of the paper.

1. In the abstract, in the first line, it is written that the copper tube prepared by this method (method name must have been disclosed first)

2. In the experimental section, the basis of the selection of parameters and their levels must have been discussed.

3. The Experimental findings presented in graphs 8-10 must not have joined with lines as they are not trends. Authors can also display an error bar if possible.

4. The conclusions can be better highlighted.

5. Very less references have been used by authors and can be increased considering the length of the article.
